# ECG Biometrics Using Deep Learning and Relative Score Threshold Classification

**DOI:** 10.3390/s20154078

**Published:** 2020-07-22

**Authors:** David Belo, Nuno Bento, Hugo Silva, Ana Fred, Hugo Gamboa

**Affiliations:** 1LIBPhys, Physics Department, Faculty of Sciences and Technology, Nova University of Lisbon, 2825-149 Caparica, Portugal; n.bento@campus.fct.unl.pt (N.B.); hgamboa@fct.unl.pt (H.G.); 2Instituto de Telecomunicacoes, Instituto Superior Tecnico (IST), Technical University of Lisbon, 1049-001 Lisboa, Portugal; hsilva@lx.it.pt (H.S.); afred@lx.it.pt (A.F.)

**Keywords:** deep learning, biometrics, electrocardiogram, convolutional neural network, recurrent neural network, authentication, identification, artificial neural networks, biosignal, RLTC

## Abstract

The field of biometrics is a pattern recognition problem, where the individual traits are coded, registered, and compared with other database records. Due to the difficulties in reproducing Electrocardiograms (ECG), their usage has been emerging in the biometric field for more secure applications. Inspired by the high performance shown by Deep Neural Networks (DNN) and to mitigate the intra-variability challenges displayed by the ECG of each individual, this work proposes two architectures to improve current results in both identification (finding the registered person from a sample) and authentication (prove that the person is whom it claims) processes: Temporal Convolutional Neural Network (TCNN) and Recurrent Neural Network (RNN). Each architecture produces a similarity score, based on the prediction error of the former and the logits given by the last, and fed to the same classifier, the Relative Score Threshold Classifier (RSTC).The robustness and applicability of these architectures were trained and tested on public databases used by literature in this context: Fantasia, MIT-BIH, and CYBHi databases. Results show that overall the TCNN outperforms the RNN achieving almost 100%, 96%, and 90% accuracy, respectively, for identification and 0.0%, 0.1%, and 2.2% equal error rate (EER) for authentication processes. When comparing to previous work, both architectures reached results beyond the state-of-the-art. Nevertheless, the improvement of these techniques, such as enriching training with extra varied data and transfer learning, may provide more robust systems with a reduced time required for validation.

## 1. Introduction

Some traditional methods of identification and authentication, such as key-possession or username-password, may prove ineffective against terrorist and criminal acts. Key areas of society, such as border control, criminal identification, and electronic transactions, demand more reliable solutions provided by Biometric Systems (BS). Accordingly, both the public and private sectors have been increasing the investment in these solutions in the last decade [1,2].

The field of biometrics is a pattern recognition problem, where the individual traits of a person are coded, stored into a database, and compared with the recorded biometric data according to two approaches: identification (finding the registered person from a sample) and verification, also known as authentication (prove that the person is whom it claims) [1,3,4]. These systems BS may be evaluated in several dimensions, such as universality, uniqueness, permanence, measurability, performance, acceptability, and circumvention [3,5].

Since the application of elementary fingerprint recognition in 1883, research has been searching for different and more reliable types of biometric data [6]. Nowadays, fingerprints, faces, hands, veins, voices, irises, and other biological and behavioral features are used to identify subjects [7,8]. An example of another potentially useful biometric signature is physiological signals. These are electric signals generated from the electrochemical changes in nerve cells, muscles, or gland cells, which may be acquired from surface electrodes in contact with the skin. Some examples of physiological signals used in biometrics are: the Electrocardiogram (ECG), extracted from the heart; the Electromyogram (EMG), from muscle activity; the Electrodermal Activity (EDA), which tracks changes in skin conductivity, expressing the activity of the sympathetic nervous system; and the Electroencephalogram (EEG) [9], extracted from brain activity [3,10].

### 1.1. ECG Biometrics

Given its morphological signature and advances in sensing devices, the ECG has emerged as a biometric modality with the promise of robustness against circumvention attacks and the ability for continuous pervasive acquisition scenarios. Notwithstanding this, using non-invasive ECG presents its challenges, such as noise, artifacts, and intra-subject variability [10,11]. The computation of features may prove even more arduous when the contamination is present in characteristic points of the waveform [12].

The sources of noise and more abrupt changes of the signal (artifacts) include electrode material, sensor location, power-line interference, movement artifacts, and instrumentation of the devices [13]. The intra-subject variability may be due to changes in the health status of the patient, heart-rate variability, physical exercise, affective end emotional states, and drug consumption [10,11]. Nevertheless, the ECG remains stable over the years, sufficiently to allow recognition with an error rate of 3% [14].

The morphology of the ECG cycle is related to its electrophysiological activity and comprises the P, PQ interval, QRS complex, ST interval, and T waves [11,14]. According to the used features, ECG biometric systems can be categorized as fiducial-based, in which heartbeat templates are composed of features related to the relevant points of each cycle, and non-fiducial-based, which consider the ECG signal as a whole, computing its features accordingly [1]. The inclusion of both types of elements, to achieve better results, defines these systems as partially-fiducial [10,15,16].

### 1.2. Objective

ECG biometric systems struggle to find their way outside controlled settings due to the intra-variability existent in real-life scenarios. Therefore, this work aims to provide one step further in the abstraction of the individualized ECG signal to be used in real-life scenarios using state-of-the-art technologies such as Deep Neural Networks (DNN).

In order to achieve this goal, two systems with distinct architectures are proposed (Figure 1) as an effort to improve performance in both identification and authentication processes without extracting human-crafted features: (1) a non-fiducial system which uses a Recurrent Neural Networks (RNN) capable synthesizing ECG signals, outputting a score based on the error of prediction; (2) a partial-fiducial system which uses a Temporal Convolutional Neural Network (TCNN), giving a score based on the output of the last layer. Both scores computed and fed to the Relative Score Threshold Classifier (RSTC), which classifies a single sample window or a chunk of windows. Both systems will be benchmarked in three public databases used in BS literature: Fantasia [17], MIT-BIH [18], and CYBHi [13,19].

## 2. Related Work

The literature that refers to the application of DNN architectures in ECG biometry is mainly comprised of feed-forward or Convolutional Neural Networks (CNN) architectures. The practice of using human-extracted features may be seen in [20] where a temporal frequency spectrograms are produced with discrete wavelet transform. The extracted features from these time-frequency matrices are fed into a feed-forward network for classification. As for the authors of [21], the feature extraction module is bypassed, using the template of the ECG cycle as input to a feed-forward network, for both feature extraction and classification. CNN architecture is also used in this context, in the works of [22,23], this architecture is explored in both authentication and authentication biometric processes. These fiducial systems feed their networks with the average cycle of the signal, based on the neighborhood of the R peak of the ECG signal. The systems developed by [24,25,26] also use DNN technologies, but since they are trained and tested with the benchmarked datasets, they will be discussed later in the ”Fantasia”, ”MIT-BIH”, and ”CYBHi” subsections.

The improvements of these methods cover the generalization of the abstract notion of an individualized ECG by employing one-dimensional time-series, as opposed to images or human-extracted features and limiting the need to extract the QRS complexes, by developing non-fiducial and semi-fiducial systems. This claim is empowered by including an off-person database which has high intra-variability in most of the subjects. For benchmarking purposes, the rest of this section will address related work using the referenced databases: Fantasia, MIT-BIH, and CYBHi.

### 2.1. Fantasia

PhysioNet was created under the auspices of the National Center for Research Resources of the National Institutes of Health [27] and comprises several public physiological databases, such as MIT-BIH and Fantasia. The Fantasia dataset contains 20 subjects aged between 21 and 34 years old and 20 aged between 68 and 81, during 120 min of continuous supine resting ECG recording while watching its homonym Disney movie [17].

This database was adopted by [28], which created a model that uses a reduced version of a set of fiducial features by applying Principal Component Analysis (PCA), Linear Discriminant Analysis (LDA), information-gain ratio, and rough sets (RS) achieving an accuracy of 90 ± 8% for PCA, False Rejection Rate (FRR) 5% for LDA, and False Negative Rate (FNR) of 4% for RS [28]. The same team published a method that extracts and decomposes R–R intervals using Discrete Wavelet Transform (DWT) and achieved 95.89% accuracy, FRR 0%, and FNR 5% [29]. The work of [15] includes the analyses of the influence of the heart rate variability in the QT intervals and suggests their correction to improve performance, leading to the identification rates with MultiLayer Perceptron (MLP) and Support Vector Machine (SVM) of 97% and 99%, respectively. The study made by [24] feeds spectrograms’ extracted features into a feed-forward network, such as [20], achieving an accuracy of 97.2% for the Fantasia dataset; this work also has a reference to the MIT-BIH database.

### 2.2. MIT-BIH

The MIT-BIH Database has been available since 1999 in PhysioNet. The basic subset of this database, MIT-BIH Arrhythmia, contains ECG records from 47 subjects with 360 Hz of sample frequency and 11-bit resolution, from Boston’s Beth Israel Hospital. More subjects, with no significant arrhythmic episodes, were added to this dataset, 18 subjects in MIT-BIH Normal Sinus and seven individuals in MIT-BIH Long-Term [18].

This database has been used not only for validating arrhythmia detectors and other cardiac dynamics analysis but also in the biometrics field. The MIT-BIH Normal Sinus Rhythm subset was adopted by [30,31,32] to validate their models. The feature extraction method proposed by [30] is comprised of Piecewise Linear Representation and Dynamic Time Warping (DTW) methods, and the computation of similarity measurements is used for classification. Overall, the identification results were close to 100% with the minimum of the half total error (HTER) of 0.2%. As for [31], the accuracy has reached 99.07% by using a combination of ten features from heartbeat and sixty from coefficients of the Hermite Polynomials Expansion fed into a Hidden Markov Model (HMM). In another case, several classification methods were applied with characteristics of the QRS complex by [32] achieving the accuracy of 98.3% for the Bayes Network, 99.07% for Naive Bayes, 99.07% for Multilayer Perceptron, and 99.07% for k-Nearest Neighbor [32].

Combined subsets of the MIT-BIH were used by [24,33]. The first uses a semi-fiducial approach in which a Wavelet Transform was applied together with an Independent Component Analysis (ICA) analysis to detect each heartbeat. This information concatenated with the R–R intervals information and fed into a SVM classifier. For the selected 23 records, which include the Normal Sinus Rhythm and the Arrhythmia subsets, this work reached 86.4% accuracy for subject evaluation, even though the focus was to classify heartbeat classes. The mentioned work from [24] achieved 90.3% of accuracy for 47 individuals of MIT-BIH. The work developed by [25] claims to achieve a 98.55 % of accuracy in identification paradigm using a bi-directional Gated Recurrent Units (GRU) network model incorporating for 47 records of the MIT-BIH Arrhythmia Database. In this article, they compare several architectures, including CNN and Long Short-Term Memory (LSTM) models with a fiducial approach.

### 2.3. CYBHi

The *Check Your Biosignals Here initiative* (CYBHi) dataset was acquired with an off-person procedure without the use of skin electrodes. This database has been considered by [13] as one of the two best public databases in terms of acquisition protocol and hardware for biometric studies, in comparison with fifteen databases. The setup of the acquisition system consists of a pair of synchronized sensors that were hand-shaped, in which the electrodes were placed on the fingers. The data consists of two 2-minute acquisitions three months apart of 63 participants, 14 males, and 49 females (18–24 years old) [19]. For this paper, the sessions will be named M1 and M2.

In literature, the outcome for authentication of [19] was an Equal Error rate (EER) of 9.1% using an SVM classifier against the correlation between features extracted from ECG cycle templates. Concerning [5], the extracted features from the ECG morphology are joined with the features extracted from the R–R intervals with Nearest Neighbor Algorithm (NNA) and used for classification using a Euclidean distance classifier. The identification rate of this study is 95.2% for the first test and 90.2% for the second test. The work of [26] uses two CNN networks are fused together, one fed with the raw ECG cycle, while the other uses the cycle’s spectrogram, described before, achieved EER values of M1 vs. M1 of 1.33%; M1 vs. M2 12.78%; M2 vs. M1 13.93%.

## 3. Materials and Methods

As stated before, two architectures were explored and compared using the same classification method, the RSTC. The setup comprises RNN and TCNN approaches without the use of human-extracted features. Even though both procedures are trained to create an internal representation of the ECG waves, they have different outputs for calculating the same similarity score. While, for the RNN approach, the score is based on the prediction error, the TCNN score is given by the output given by the last layer, the logits layer. Before training and testing models, heavily corrupted windows were removed by using the algorithm developed by [11] using the standard deviation feature.

Cross-validation was used for training and testing. For Fantasia and MIT-BIH, the training was made within the first 33% of the ECG records, while the rest was reserved for the testing set. As for the CYBHi database, the training and testing sets were divided for 50% of the dataset when each session was tested, while, when crossing both sessions, 100% of the respective session was used. More details will be given in the Results section. All of the inter-sectioned windows were removed to ensure that both datasets did not share information between them.

### 3.1. Recurrent Neural Network Approach

The synthesizing of ECG signals in the previous work [34] provided a hypothesis that, if the model is trained with the signal from subject A and fed with the signal from subject B, the prediction error will be higher than if fed with the signal from the original subject A. Subsequent to the observation of the results while testing this premiss, it was confirmed that it could be further explored. The hyper-parameter tuning was made through careful observation and continuous try-and-error procedures in search of better results using a subset of the Fantasia dataset.

In the RNN approach, data are firstly pre-processed (Figure 2), and later transformed by an embedded matrix—*E*, three sequential GRUs—*G*, a dense layer with linear activation and a softmax node [34].

### 3.2. Pre-Processing

The pre-processing step starts with removing the moving average, followed by a convolution with a *Hanning* window and normalized by a moving absolute maximum window. The benefit of this approach is to mitigate the unwanted low and high frequencies resultant from the signal acquisition. Following this procedure, the edges of the signal’s amplitude histogram were clipped, according to a value of confidence, to reduce the influence of artifacts. This value was typically 0.5% but could increase depending on the level of corruption of the signal. Due to the nature of the GRU networks, the quantization of the signal (*s*) must be performed. This step consists of reducing the dimension of the possible amplitude values. After the transformation, the signal will be comprised of $S_{D}$ (signal dimension) number of possible values, which was set to 512. The higher the value of the signal dimension ($S_{D}$), the higher the detail is retained from the signal. In summary, the resultant signal (*x*) is obtained with the following equation:(1)$$x_{n}=\mathrm{round}\left(\left[\frac{s_{n}-min\left(s\right)}{max(s-min(s))}\right]\;\cdot \;\left[S_{D}-1\right]\right)\;,$$
where $x_{n}$ is the *n*-th sample of the input vector and $s_{n}$ is the raw *n*-th sample of the raw signal (*s*). The variable $k\in \{0,1,\ldots S_{D}-1\}$ represents the number of possible integer values $x_{n}$ may take.

Finally, the signal was segmented into windows of size *W* with overlap between them. The overlap percentage was 67% for the “Fantasia” and “MIT-BIH” datasets and 89.1% for the “CYBHi” datasets, to increase the amount of data.

### 3.3. Architecture

The architecture depicted in Figure 3a comprises one embedding matrix (*E*), commonly used for text processing. It functions as a translation mechanism between the input and the three GRU layers (*G*). The input is an integer that maps to the corresponding $x_{n}$-th column of the matrix *E*. This matrix initialized randomly, but it is optimized during the training process, adjusting itself to the possible input values along with the other parameters.

As stated before, in [34], this architecture is shown to synthesize biosignals, validated by the prediction error. As the signal which trained the model is fed into itself, the prediction error is lower in contrast to the signals with other origins. In the case of biometry, this notion is expanded to the specific source of the signal, i.e., this network should be able to learn the ECG individual intricacies of each subject. Following this reasoning, the produced similarity score (S(p,i,w)) is given by the following equation:(2)$$S\left(p,i,w\right)=\frac{e(p,i,w)}{max(e(:,i,w))}\;,$$
where *e* is the prediction error made by each predictor—*p*—for the time-window *w* of the signal which belongs to the subject *i*. The expression max(e(:,i,w)) represents the maximum for the same window *w* for all predictors. Each model is trained for each subject while predicting the next sample amplitude. These models were trained using mean-squared error using the RMSProp [35] optimization algorithm. The prediction error is provided by the loss function given by the following equation: (3)$$e_{p}=\frac{1}{W}\sum_{n=0}^{W-1}{\left(\hat{y_{n}}-y_{n}\right)}^{2}\;,$$
where the *W* is the window size (number of samples in the time-window), ${\hat{y}}_{n}$ the predicted *n*-th sample and $y_{n}$ the real sample value. Note that the $s_{n}$ is the raw signal, $x_{n}$ is the quantized signal, while $y_{n}$ is equal to $x_{n}$ dephased by one sample since the RNN architecture predicts the next sample [34].

In summary, the parameters required for this architecture are the time-window size (*W*), the overlap ratio, the edges cropping ratio, the quantization dimension of the signal ($S_{D}$), and the number of hidden nodes in each GRU ($H_{D}$).

### 3.4. Temporal Convolutional Neural Network

The second proposed approach is a two-stream TCNN [36], which uses one-dimensional convolutional layers with dilated convolutions to learn temporal patterns. It combines predictions from two different inputs: ECG non-fiducial window segment and a sub-segment of the same window containing only one full cycle centered in R peak of the QRS complex. Earlier configurations of this CNN were assembled and experimented, such as including spectrograms as input [37], and others including a one-dimensional CNN without the use of dilated convolutions. After testing using Fantasia and CYBHi datasets with try-and-error parameter hyper-tuning, the resultant architecture provided the best results in relation to the previous versions.

Before data are fed into the networks, the signals are pre-processed by removing the moving average, convolution with the *Hanning* window, normalized with the absolute maximum, and finally segmented, analogous with the RNN procedure.

Within this architecture, depicted in Figure 3b, each network is composed of several convolutional layers with 24 kernels of size 4, followed by batch normalization and a ReLU activation function. At the end of each network, two fully connected layers, the first with 256 units while the second with the number of individuals are applied to the end of each network. The number of convolutional layers for the Fantasia and MIT-BIH databases is 6 of sizes 256, 128, 64, 16, 8, and 4 for the non-fiducial network and 4 of 64, 16, 8, and 4 for the fiducial network. The input window had 256 samples for these databases. For the CYBHi database, due to the increase of the sampling frequency, each network had an increase of two initial layers, 1024 and 512 added to the previous layers of the non-fiducial network, and 256 and 128 added to the previous layers of the fiducial network. The input window of this dataset had 1028 samples.

Each CNN was trained using the Adam optimizer [38] with the cross-entropy loss function (*L*), given by:(4)$$L=-\frac{1}{I}\sum_{i}^{I}\left[i\log\left(\hat{i}\right)+\left(1-i\right)\log\left(i-\hat{i}\right)\right]\;,$$
where the *I* is the number of subjects, $\hat{i}$ is the predicted subject index, while *i* the real subject index. Subsequently, after each network is trained independently, the logit vectors from both networks are fused with the sum rule. The score, in this case, is given by subtracting one to the normalized output vector (*o*), according to this equation:(5)$$S\left(p,i,w\right)=1-\frac{o(p,i,w)}{max(o(:,i,w))}$$

The parameters specific for this architecture are the time-window size (*W*), the overlap ratio, the size, and the number of filters in each of the convolutional layers, and the size of the fully connected layer before the logit layer. The number of layers depends on the size of the input window.

### 3.5. Relative Score Threshold Classification

The RSTC is a simple method that classifies by choosing the lowest normalized similarity score ($\bar{S}\left(p,i,b\right)$) for each batch of windows (*b*); consequently, *S* is a three-dimensional tensor with dimensions (I×I×B), where *B* is the size of the batch. This measurement is calculated after normalizing in relation to each predictor. When *B* is higher than one, then only the minimum of that batch of windows is considered, according to the following function:(6)$${\tilde{S}}_{p,i,b}=\min\left[\left\{S_{p,i,B\;\cdot \;b},S_{p,i,B\;\cdot \;b+1},\ldots ,S_{p,i,B\;\cdot \;b+B}\right\}\right]\;,$$
where the notation B·b represents the multiplication of the batch number by the batch size, for index purposes. The following normalization for each *b* in relation to the *p* is determined by:(7)$${\bar{S}}_{p,i,b}=\frac{{\tilde{S}}_{p,i,b}-\min\left({\tilde{S}}_{:,i,b}\right)}{\max\left[{\tilde{S}}_{:,i,b}-\min\left({\tilde{S}}_{:,i,b}\right)\right]}\;,$$
where minimum and maximum values are calculated in respect to all predictors for each batch of windows. Considering that $\bar{S}\left(p,i,b\right)\in \left[0,1\right]$, the value 0 encodes to the most probable class, while 1 codes to the lower probable one. The predicted class for the identification paradigm ($C^{I}$) for each individual (*i*) and batch of windows (*b*) is given by:(8)$$C_{i,b}^{I}=\underset{p}{arg min}\left[{\bar{S}}_{:,i,b}\right]$$

### 3.6. Evaluation

The identification evaluation made by the multimodal classifier ($C^{I}$) is given by the accuracy, specificity, and sensibility. The authentication evaluation is made by the EER that is obtained on the Receiver Operating Characteristic (ROC) curve, when FNR = False Positive Rate (FPR) for each binary classifier ($C^{A}$). The authentication paradigm is evaluated with this measurement because these systems prioritize the compromise between the minimization of the possible imposters to breach the system, with the maximization of the number of rightful individuals who can access the system. Each binary classifier is made by comparing the score of all predictors *p* for each individual *i* and all windows *w* with a changing threshold (δ). This system is fine-tuned for each individual and the output gives if that specific *b* of windows belongs to the claimed individual:(9)$$C_{p,i,b}^{A}=\left\{\begin{array}{cc} 1, & if\;\bar{S}\left(p,i,b\right)\leq \delta \\ 0, & if\;\bar{S}\left(p,i,b\right)>\delta \end{array}\right.$$
When $C_{p,i,b}^{A}$ has the value 1, the classification for class *A* is positive, while 0 when negative. The ROC is calculated with an array of changing thresholds and the EER may be extracted for all the intersections with the diagonal where FNR = FPR. Since each individual has its own fine-tuned ROC, the mean and standard deviation must be calculated and used for evaluation. An example of how these thresholds are obtained is depicted in Figure 4

## 4. Results

This section will present the results according to each database, ordered in increasing average intra-variability across subjects. Apart from the validation for identification and authentication processes, the Fantasia dataset will be the benchmark to test the used methods and robustness of the proposed algorithms.

### 4.1. Fantasia

All data were segmented in windows of 512 samples (approximately 2s) with an overlap of 67%. After the rejection algorithm, the selected windows were on average 1464 windows of a total of 3787 per subject.

The score distribution per subject when the test set is fed into the RNN model trained with ECG,8 is depicted in Figure 4. The scores distribution, which is presented in Figure 4a, considers all batches of windows with B=1 and B=20 in Figure 4b. When analyzing this example, it can be observed that, with the increase in the number of windows contained in each batch, both average and standard deviation are reduced. The three thresholds presented in Figure 4 will have a different binary classification for the predicted ECG,8 when fed to different values of *B*. In the case of (a), the $\delta_{1}$ threshold will give a positive value for most of the ECG23 batches, which is false. When both $\delta_{2}$ and $\delta_{3}$ thresholds are considered, most of the ECG,8 batches will increase, at the cost of increasing also the number of imposters.

When the *B* increases to 20, the variance and average value of ECG8 will decrease significantly. In this case, $\delta_{1}$ will now classify correctly most, or even all, of the ECG8 batches, while reducing the number of ECG23 classified incorrectly, ensuring lower values for the FNR and FPR. Both $\delta_{2}$ and $\delta_{3}$ thresholds will produce more imposters while maintaining the same number of correct classifications. The ROC curve is generated for each predictor and each batch size, producing different values for EER that can be interpreted for authentication systems.

The identification rate, i.e., accuracy for the identification paradigm, for both algorithms is depicted in Figure 5. Close inspection reveals that both algorithms increase accuracy with time per batch, but the TCNN algorithm starts with higher accuracy, being outperformed by RNN after approximately one minute of signal per batch. While the RNN algorithm increases until reaching close to 100% at approximately 112 s, the TCNN algorithm reaches 99.1% after approximately 90 s, but its curve displays a steady behavior. The best results are presented in the confusion matrices of Figure 5b for RNN and Figure 5c TCNN approaches.

The results for the authentication mode are depicted in Figure 6a,b which show the evolution of the EER per time contained each batch. Both RNN and TCNN achieve values very close to 0% at 80 s, but the TCNN reaches those values with a lower standard deviation and sooner than the RNN counterpart.

### 4.2. MIT-BIH

All the MIT-BIH signals were resampled to a frequency of 250 Hz to ensure that the same sampling frequency is maintained in all datasets for a rigorous comparison. The rejection rate of the noise removal algorithm was on average 42% for each subject.

The conducted identification test utilizing the MIT-BIH database can be observed in Figure 7, showing that the accuracy curves have similar behavior as the ones in the Fantasia dataset. For the sake of comparison with the bibliography, an analysis of the Normal Sinus Rhythm dataset was also made in detail (Figure 7). The RNN approach does not go above 93% for all the MIT-BIH but reached almost 100% accuracy for Normal Sinus Rhythm. When analyzing Figure 7b, it is possible to see the diagonal clearly, but some trained models generalize the ECG signals instead of the individual traits and, consequently, classifying incorrectly. Some possible limitations include the loss of information during resampling, sensibility to noise, and symptomatic behavior of the arrhythmia events. The last option is supported with the results close to 100% accuracy for the Normal Sinus Rhythm subset.

Concerning the TCNN approach, even though the 96.4% is reached after more than 2 min for the full MIT-BIH database, one can observe that values close to 96% are reached after 10 s, reaffirming its stability and robustness to noise. The confusion matrix depicted in Figure 7c shows that only two predictors display a faulty behavior, probably due to the presence of arrhythmia events. When examining the individuals from the Normal Sinus Rhythm, the accuracy reached almost 100% after approximately 10 s (Figure 7a).

The authentication evaluation of all the MIT-BIH individuals is displayed in Figure 8a,b. These figures show that the TCNN approach reaches EER values close to 0% with 72 s in each batch and approximately 1.5% for RNN. Even though the best result for TCNN is after 1 min, the values are quite low from the start.

#### CYBHi

As mentioned before, the CYBHi database comprises two sessions with a time distance of three months. Therefore, both algorithms are evaluated with all the combinations of both sessions (M1 and M2) for cross-validation, simulating the enrolment and verification environments of a biometric system. This means that each model was trained using the first element and tested with the second of the following sets: “M1 vs. M1”, “M2 vs. M2”, “M1 vs. M2” and “M2 vs. M1”.

The segmentation of the signal was made with an 89% of overlap to increase the number of available time-windows. The clustering method rejected 8.93 ± 22.2% of the total signal length on the first session and 9.97 ± 24.2% on the second. As stated before, when the training and testing session was the same (“M1 vs. M1”, “M2 vs. M2”), the data were separated by 50% and 100% of each session across sessions (“M1 vs. M2” and “M2 vs. M1”).

Since this dataset displays a high variability within subjects due to the low Signal-to-Noise Ratio (SNR), the robustness of the method is imperative for good results. Figure 9a shows that the TCNN algorithm achieves results above 90% for “M1 vs. M1” and 100 % for “M2 vs. M2”, even if the amount of information was significantly lower than the previous datasets. The decrease of crossing both sessions was significant as expected, revealing accuracies only above 75 % for both “M1 vs. M2” and “M2 vs. M1”. The low accuracy presented by the RNN approach reflects the sensitivity to noise. Figure 9b shows that the EER achieved 4.3% and 6.1% for “M1 vs. M1” and “M2 vs. M2”, respectively, using the RNN models, while TCNN displays results close to 0%, probably due to the temporal proximity of the training and testing sets. As for the evaluation across different sessions, the EER for the RNN approach is 18.3% and 18.8% for “M1 vs. M2” and “M2 vs. M1”, respectively, and 2.2% and 4.1% for the TCNN in the same order. These results outperform the current state-of-the-art for off-person measurements.

## 5. Discussion

Table 1 summarizes the results per database concerning the consulted literature. This report suggests that the TCNN approach outperforms most of the other studies in both identification and authentication paradigms. The robustness of this architecture is mainly due to the combination of bringing together both fiducial and non-fiducial machine-learned characteristics of the signal through the fusion layer. The only exception was for the ”CYBHi M1 vs. M2” results for the identification modality, as Lourenço et al. (2012) [5] has better results, which can be explained mainly due to two reasons: (1) only 32, as opposed to 68, people were used and (2) the procedure was completely fiducially-based. The high accuracy that both approaches obtained comes with a cost of training time. For example, the RNN training time, using the CYBHi database and a computer with an Nvidia GTX 1080Ti GPU, had an average of 36 hours per subject, totaling approximately 2300 hours for this database, but only takes a few seconds to print the testing results. For the implementation of this system at an industrial level, this limitation should be mitigated as retraining should be faster, especially when the number of individuals increases.

It is possible to observe that the results for the CYBHi dataset regarding inter-sessions are lower due not only to the increase of the noise and artifacts displayed in the signal but also to the changes in the mental and emotional states of each user. This statement emphasizes the challenge that the biometrics systems face due to the intra-variability of the subjects. Hence, more acquisition setups should be made, such as expanding the number of people, extending the acquisition time, and increasing the number of acquisition sessions. Added to this, the amount of cleaner data would impact significantly the training of the models and, consequently, these measures would not only provide higher accuracy and EER but also more confidence in the results shown to claim that the systems were capable of a successful generalization.

The implementation of a biometric system including the TCNN approach in a real-life scenario recognizes several challenges: (1) the high computational time, which increases significantly with the number of samples per window and the number of individuals; (2) when a new person is added to the model, all training must be repeated; (3) this approach is viable for a suitable number of individuals, but to use on a scale of millions, the training and testing process will be costly or even impossible with state-of-the-art computational power and memory; and (4) the need to extract the peak information, which sometimes is not feasible, especially when the slow waves (such as T or U) reach higher values than the R wave.

For tackling these concerns, the use of transfer learning could prove useful. If the training of a network introduced a first learning stage with a different database for recognition of the basic structures and inner mechanisms of the ECG morphology, the second stage of learning, it would fine-tune the appropriate filters for source recognition and optimize faster. Data augmentation would also be a solution to improve the training procedure, but with the intent to avoid over-fitting of the training data, novel solutions should be addressed. The current ECG synthesis methods need to encompass the variability of the data, but to keep certain aspects of the frequency due to the non-fiducial aspects of these algorithms. The study conducted by Belo et al. (2017) [34] shows great promise for this endeavor, but experiments have shown that the frequency of the synthetic ECG was very machine-like in terms of frequency, as it did not embed the semi-randomness of the cyclic behavior of the signal.

Even though the RNN achieves state-of-the-art results for the Fantasia and Sinus Normal Rhythm of the MIT-BIH, the sensibility to changes in the signal is higher than the aforementioned architecture. The main advantages of the RNN architecture are that all the models are independent, so there is no need to train them all again when adding a new person to the biometric system, the fact that it is non-fiducial, this system is viable for good quality acquisitions, and it is possible to generate the signal, in order to track if the model is well trained.

## 6. Conclusions

This study has shown the usefulness of DNN in enhancing current biometric systems. These technologies allied with the massive collection and storage of biological data will provide powerful and robust systems, increasing the level of security, reducing the concerns of counterfeiting actions. Even though the challenges presented by the intra-variability issue impacted the results of these architectures, it is possible to state that they improved comparing with the literature increasing one step towards the true abstraction of each subject’s ECG signal.

Even though the RNN and TCNN systems provided good results, their limitations could be surpassed if both architectures were combined. Other state-of-the-art downsampling methods could also be applied to mitigate the loss of information while increasing the training speed. The next steps to implement in a real-life scenario would be the introduction of transfer learning in the training process, the acquisition of extra time windows per person, the addition of more individuals, and the extension of the number of sessions through time. The study of the impact of the noise and the scalability of these systems would also give an added value. Finally, the training/retraining/inference time of these systems should be analyzed and enhanced for real-life applications for delivering close to real-time solutions.

One factor that could give a higher generalization and detection capabilities would be the inclusion of different electrode configurations for the clinical-grade ECG so that the DNN model learns the different angles of the signal gaining robustness over the different placements. The analysis of different emotional and mental states could also be very important, as the ECG signal can change under different conditions.

## Figures and Tables

**Figure 1 sensors-20-04078-f001:**
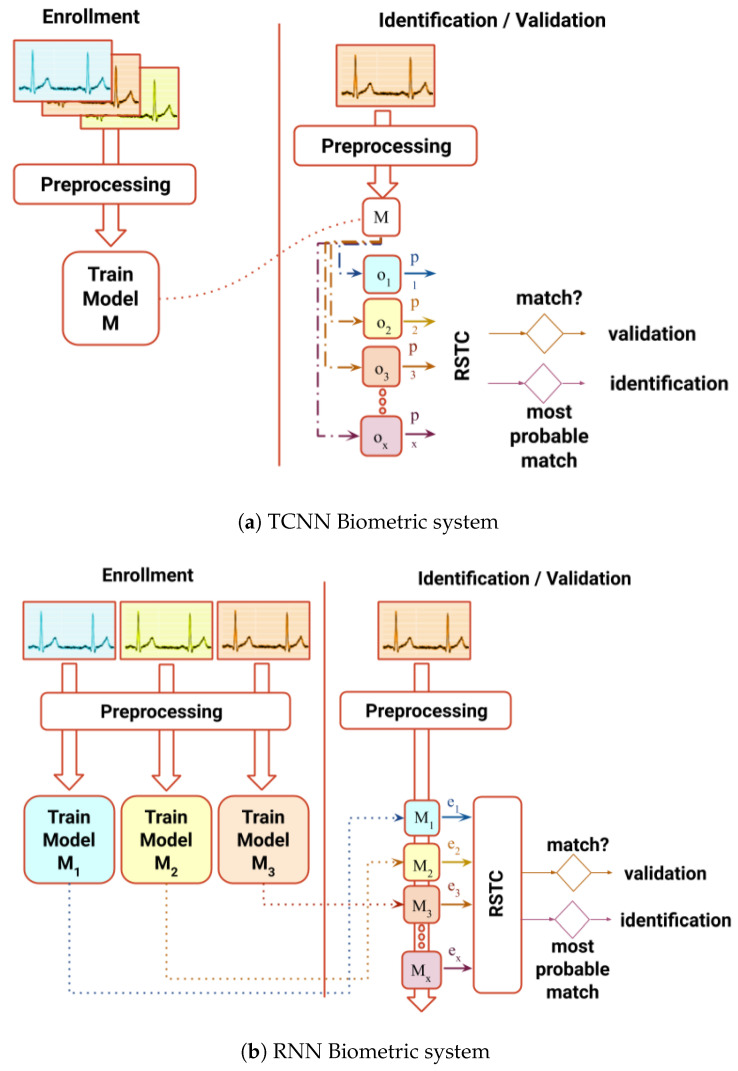
Proposed biometric systems. In (**a**) the trained model extracts features and has an output which is used to calculate the probabilities—p—for each class, as for (**b**) the error of prediction—e—of each model is used. RSTC is the classifier that validates or identifies which person does the input sample belong.

**Figure 2 sensors-20-04078-f002:**
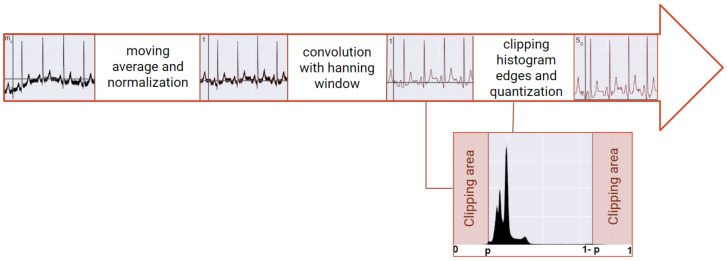
Pre-processing phase for the RNN procedure.

**Figure 3 sensors-20-04078-f003:**
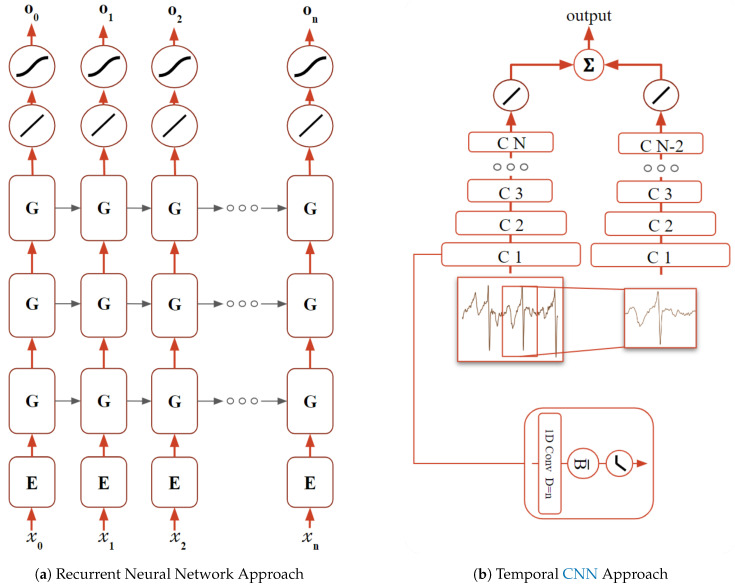
Proposed architectures. (**a**) depicts the whereRNN approach, where each input ($x_{n}$) is fed into an embedded matrix (*E*) and, consequently, into a sequence of three GRU (*G*) cells. The output of this sequence is transformed through a fully connected network and a softmax function resulting in $o_{n}$. (**b**) depicts the TCNN approach, where a time window (TW) and the first complete cycle are input to two different sequences of CNN blocks (*CN*) each comprised of a one-dimensional convolutional layer with an *N* dilation factor, batch normalization, and a ReLU activation function. One of the networks has an *N* number of blocks while, for the other *N* — 2, they both end in a fully connected network. Finally, both of the resultant vectors are summed producing the logit vector for classification.

**Figure 4 sensors-20-04078-f004:**
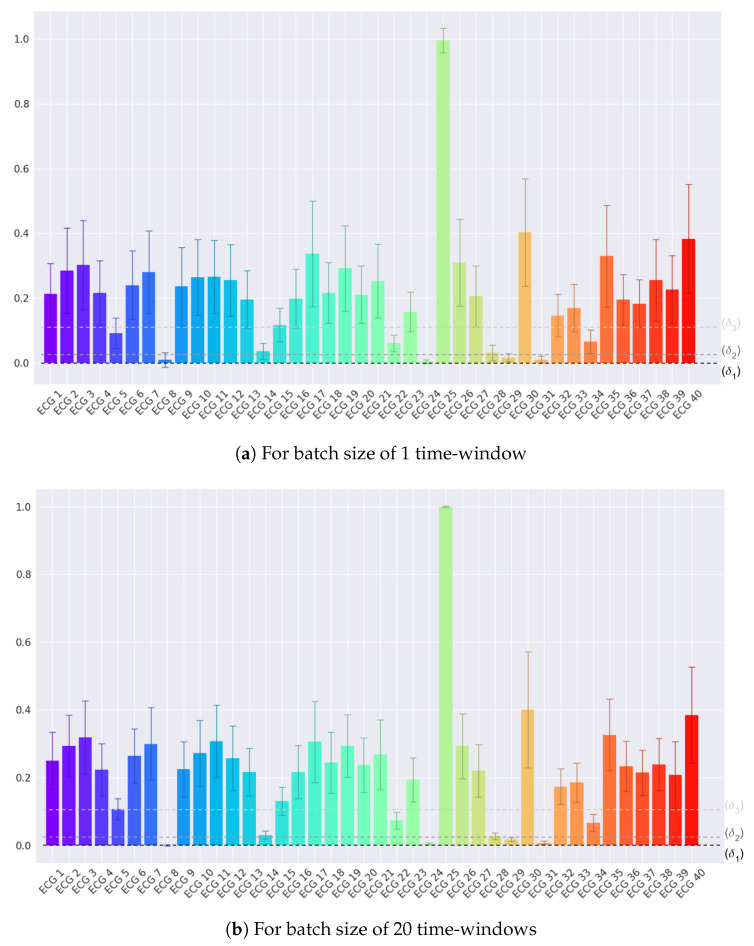
Distribution of the score of the RNN network for the predictor ECG,8 for each of the ECG signals of the Fantasia Dataset. While (**a**) refers to a single window of 512 samples, (**b**) is using the evaluation for a batch size of 20 windows. $\delta_{1}$, $\delta_{2}$, and $\delta_{3}$ are three examples of increasing threshold values.

**Figure 5 sensors-20-04078-f005:**
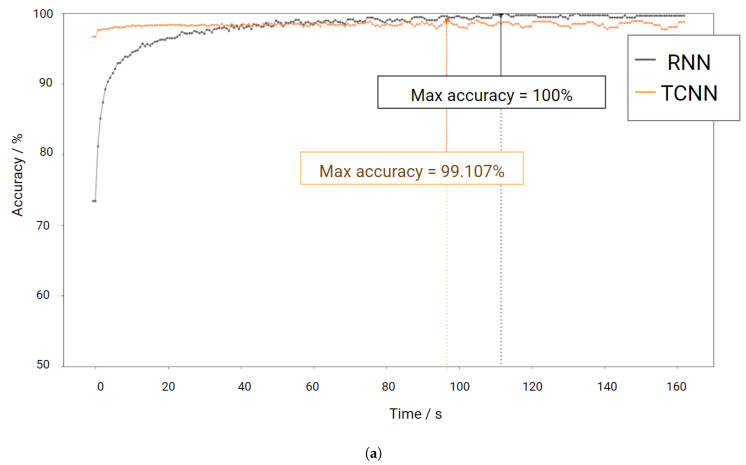
Results for the identification paradigm related to the Fantasia dataset. (**a**) displays the evolution of accuracy for the Fantasia Dataset with increasing batch size. (**b**,**c**) depict the confusion matrices for the best identification performance for both architecture approaches (ac is accuracy, sc is specificity, and st is sensitivity).

**Figure 6 sensors-20-04078-f006:**
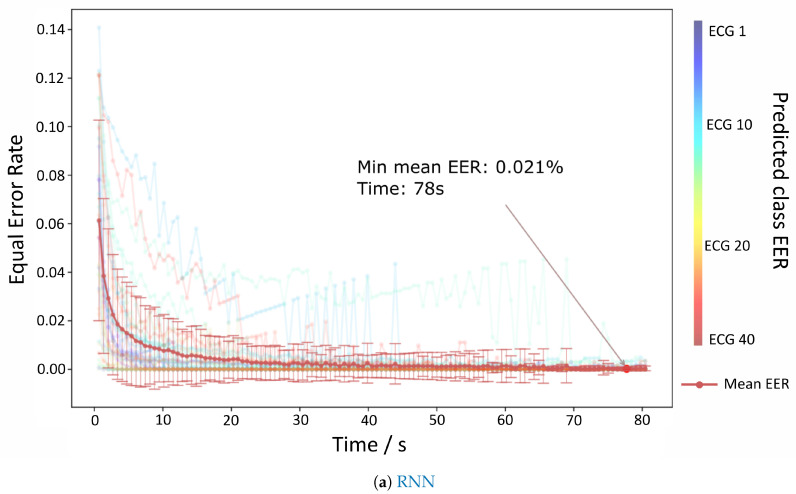
Results for the authentication paradigm related to the Fantasia dataset. (**a**,**b**) depict the evolution of the EER values over time for both approaches.

**Figure 7 sensors-20-04078-f007:**
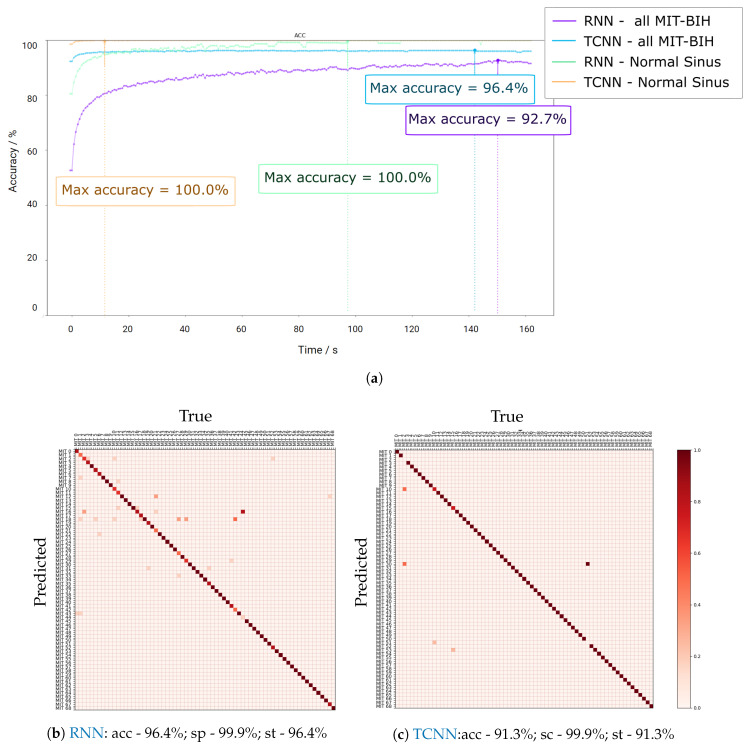
Results for the identification paradigm related to the Fantasia dataset. (**a**) displays the evolution of accuracy for Fantasia Dataset with increasing batch size. (**a**,**b**) depict the confusion matrices for the best identification performance for both architecture approaches (ac is accuracy, sc is specificity, and st is sensitivity).

**Figure 8 sensors-20-04078-f008:**
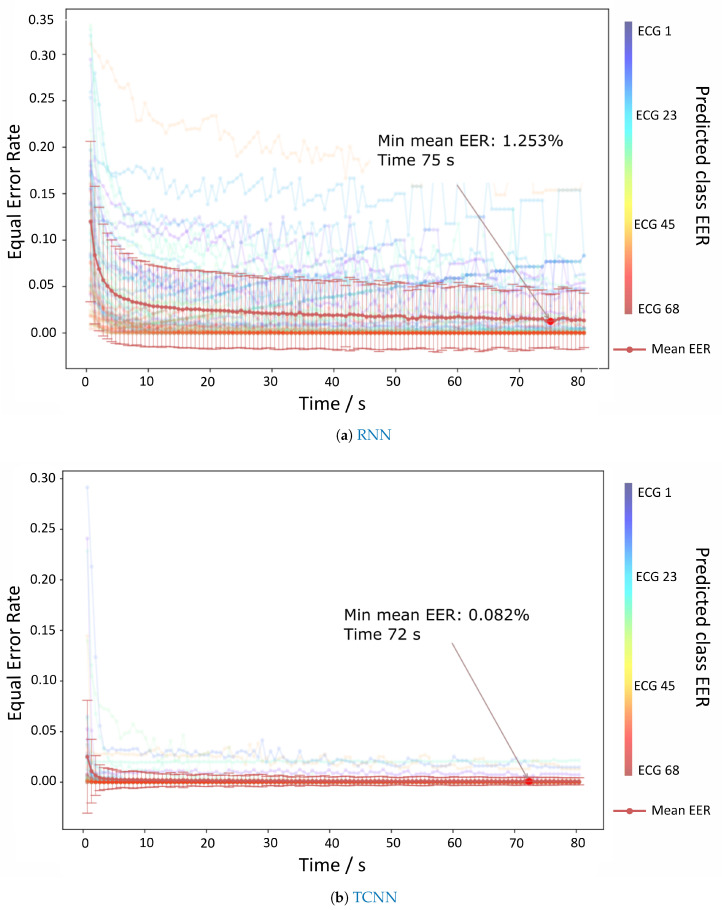
Results for the authentication paradigm related to the MIT-BIH dataset. (**a**,**b**) depict the evolution of the EER values over time for both approaches.

**Figure 9 sensors-20-04078-f009:**
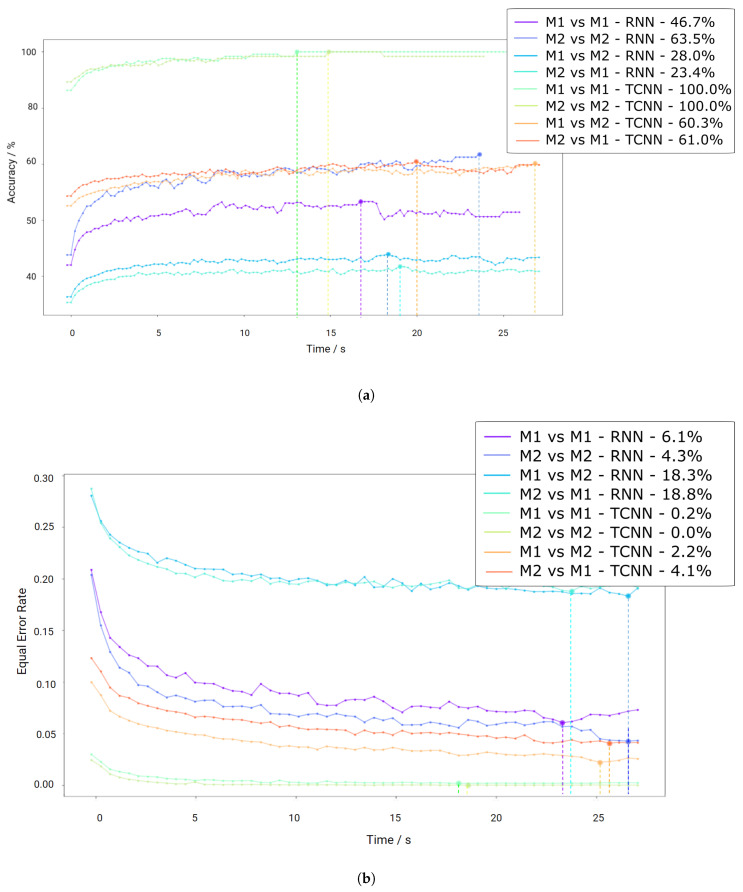
Results for the CYBHi dataset for each approach. M1 represents the first session while M2 represents the second session. (**a**) depicts the results for the identification paradigm, displaying the evolution of accuracy with increasing batch size. (**b**) displays the evolution of EER with increasing batch size for the authentication paradigm.

**Table 1 sensors-20-04078-t001:** Fantasia Dataset.

Study	Database	Acc	EER
Tantawi et al. (2013) [28]		90.0%	----
Tantawi et al. (2013) [29]		95.9%	----
Gargiulo et al. (2015) [15]	Fantasia	99.0%	----
RNN		**100%**	**0.02%**
TCNN		99.1%	**0.02%**
Wang et al. (2008) [39]		98.1%	----
Fatemian and Hatzinakos (2009) [40]		99.6%	----
Wang et al. (2008) [39]		94.5%	----
Rabhi and Lachiri (2013) [31]	MIT-BIH Sinus	99.0%	----
Sidek et al. (2014) [32]		99.1%	----
RNN		**100%**	0.6%
TCNN		**100%**	**0.0%**
Lynn et al. (2019) [25]	MIT-BIH Arr	98.6%	----
RNN	MIT-BIH	92.7%	1.5%
TCNN	Arr, Sin, Long	**96.3%**	**0.1%**
da Silva et al. (2014) [19]		94.4%	----
Lourenço et al. (2012) [5]		95.2%	----
da Silva Luz et al. (2018) [26]	CYBHi M1 vs. M1	-----	1.3%
RNN		46.7%	6.1%
TCNN		**100%**	**0.2%**
Lourenço et al. (2012) [5]		**90.2%**	----
da Silva Luz et al. (2018) [26]	CYBHi	-----	12.8%
RNN	M1 vs. M2	28.0%	18.3%
TCNN		60.3%	**2.2%**
da Silva Luz et al. (2018) [26]		-----	14.0%
RNN	CYBHi M2 vs. M1	23.5%	18.8%
TCNN		**61.0%**	**4.1%**
RNN	CYBHi	63.5%	4.3%
TCNN	M2 vs. M2	**100%**	**0.0%**

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
