# Peer review of "ECG Biometrics Using Deep Learning and Relative Score Threshold Classification"

_sensors, 2020, doi:10.3390/s20154078_

Round 1
Reviewer 1 Report
The authors present a detailed study on use of TCNN and RNN for identification and authentication of individuals using three different datasets.
The literature with regards to ECG biometrics is well cited and benchmarked.
Authors quote that their TCNN model outperforms the RNN model achieving 100%, 96% and 90% accuracy, respectively, for identification and 0.0%, 0.1% and 2.2% equal error rate for authentication for the 3 datasets.
However, considering all the datasets contain a very small sample of participants (< 100) and potentially the model that authors propose have huge number of parameters (can authors provide a number?), the variability in the model is of considerable concern. While authors do use cross-validation, the data leakage cannot be ruled out as the authors to my understanding do not use an untouched test set due to small sample size.
Data augmentation could be one technique that could be used by authors to increase the sample size. Further, for any model that has very sample size but many more fitting parameters, providing a confidence interval to the final estimates is very important in gauging the reliability. Variation across cross-validation is one way, another could be boot-strapping, to check the confidence in their estimates of accuracy and error.
Further, the intra-subject variability is another important concern as is evident from the CYBHi database where authors achieve close to 60% accuracy. This needs to be clearly highlighted as a cause of concern for robust biometric systems.
Also, while biometrics based on ECG are a candidate, I think a couple of sentences in the text should be mentioned of other potential techniques, like fingerprinting, retina scan, etc. and how biometrics could if at all be more advantageous.
A final comment I would like to make is for authors to clarify the architectures highlighted in 3(a), (b). How many parameters? what's the size of each CNN layer, what's the input dimension of images that are fed, etc. Is N=24 in their figure, etc.?
Overall, I think there is significant scope of improvement in paper as suggested above before it can be published.
Minor corrections (for example):
1. "Both outputs were 7 to the same classifier, the Relative Score Threshold Classifier (RSTC) which exploits the error of 8 prediction of the former and the logits given by the last. "
This sentence is not very clear. Can the authors rephrase?
2. "Both scores are submitted the Relative Score Threshold Classifier (RSTC)"
Missing preposition "to", also perhaps wrong usage of word "submitted", maybe fed into is more appropriate?
3. both methods will be tested "in" three?
should it be with?
4. "Recently, research in ECG biometry has been using DNN"
5. "The main advantages of the RNN architecture are the is that all the models a"
and many more ...
Author Response
The authors present a detailed study on use of TCNN and RNN for identification and authentication of individuals using three different datasets.
The literature with regards to ECG biometrics is well cited and benchmarked.
Authors quote that their TCNN model outperforms the RNN model achieving 100%, 96% and 90% accuracy, respectively, for identification and 0.0%, 0.1% and 2.2% equal error rate for authentication for the 3 datasets.
However, considering all the datasets contain a very small sample of participants (< 100) and potentially the model that authors propose have huge number of parameters (can authors provide a number?),
The number of parameters was now made clear in the section of "Methods and Materials". We agree that the variability may be an issue, but at the same time, each system can be personalized, while fine-tuning it with their own database while twerking these parameters.
the variability in the model is of considerable concern. While authors do use cross-validation, the data leakage cannot be ruled out as the authors to my understanding do not use an untouched test set due to small sample size.
Thank you for pointing out this issue, as while writing, we did think it was given, but due to your observation, we understood that it was not clear at all. During the experiments, the training and testing datasets were carefully separated to make sure that both were not mixed together. Therefore the statement “All the inter-sectioned windows were removed to ensure that both datasets did not share information between them.” was added in the introduction of the “Materials and Methods” section.
Data augmentation could be one technique that could be used by authors to increase the sample size.
The authors completely agree, unfortunately there is not many synthesis algorithms that could provide the needed variability that we required, specially frequency based (as this is a non-fiducial and semi-fiducial system). In a study we conducted, the synthesis of ECG were very machine-like in terms of frequency, as it not embedded the semi-randomness of the cyclic behaviour of the signal as it was fixed and the Poincaré for the R-R intervals displayed a cross, instead of the traditional oval shape. Therefore, we thought about using the ECG synthesis from (Belo et. al. 2017), which we mentioned, but we chose not to due to over-fitting of the training data. Based on this statement we did actually improved the “Discussion” based on this statement: “Data augmentation would also be a solution to improve the training procedure, but with the intent to avoid over-fitting of the training data, novel solutions should be addressed. The current ECG synthesis methods need to encompass the variability of the data, but to keep certain aspects of the frequency due to the non-fiducial aspects of these algorithms. The study conducted by Belo et al. (2017) [30] shows great promise to this endeavor, but experiments showed that the frequency of the synthetic ECG was very machine-like in terms of frequency, as it not embedded the semi-randomness of the cyclic behavior of the signal.” (third paragraph)
Further, for any model that has very sample size but many more fitting parameters, providing a confidence interval to the final estimates is very important in gauging the reliability. Variation across cross-validation is one way, another could be boot-strapping, to check the confidence in their estimates of accuracy and error.
The authors completely agree, some attempts were made, but unfortunately, the computational power was far too low for changes in the variation of the cross-validation and boot-strapping. Each measurement usually takes weeks to make (due to training), and some variations could take months to print the results. Consequently, most of the computations were focused on fine-tuning the architectures, pre-processing techniques, and calculation of the equal-error-rate. We deeply regret to mention that it is impossible for us to deliver a confidence level in a timely manner.
Further, the intra-subject variability is another important concern as is evident from the CYBHi database where authors achieve close to 60% accuracy. This needs to be clearly highlighted as a cause of concern for robust biometric systems.
The authors agree strongly with this statement and therefore made changes in three sections of the article: “Introduction”, “Discussion” and “Conclusion” sections to reinforce this statement.
Introduction second paragraph (which mentions the types of noise and the difficulties that the biometrics challenge on this account):
“Given its individual morphological signature and advances in sensing devices, the ECG emerged in the last decade as a biometric modality with the promise of robustness against circumvention attacks, and the ability to continuous pervasive acquisition scenarios. The main difficulties that involve the non-invasive ECG measurements are the intra-subject variability, artifacts, and noise [10,11] making the computation of the features arduous, particularly when the ECG signal is contaminated in the characteristic points of the waveform [12]. Merone et al. (2017)[13] compiled several sources of artifacts and noise, such as electrode material, sensor locations, power-line interference, movement artifacts, and instrumentation of the devices. As for the intra-subject variability, it may reside in the health status of the patient, the heart-rate variability, physical exercise, affective status, and drugs, but Wübbeler et al. (2007)[14] states that the ECG remains stable over the years, the sufficient to allow recognition with an error rate of 3% [10,11,14].“
Discussion (in which reinforces this challenge):
“It is possible to observe that the CYBHi results clearly are lower due, not only to the increase of the noise and artifacts displayed in the signal but also to the changes in the mental and emotional states of each user. This statement enforces the challenge that the biometrics systems face due to the intra-variability of the subjects.”
Conclusion (in which the statement that this work mitigates this issue)
“Even though the challenges presented by the intra-variability issue impacted the results of these architectures, it is possible to state that they improved comparing with the literature increasing one step towards the true abstraction of each subject’s ECG signal.”
Also, while biometrics based on ECG are a candidate, I think a couple of sentences in the text should be mentioned of other potential techniques, like fingerprinting, retina scan, etc. and how biometrics could if at all be more advantageous.
We added two paragraphs in the introduction with more information focused on general biometrics.
A final comment I would like to make is for authors to clarify the architectures highlighted in 3(a), (b).
- How many parameters?
Added to the TCNN architecture description: “In sum, the parameters required for this architecture are the time-window size ($w$), the ovelap ratio, the edges cropping ratio, the quantization dimension of the signal ($S_D$) and the number of hidden nodes in each \gls{GRU} ($H_D$).”
Added to the GRU architecture description: “As for the \gls{TCNN}, the parameters specific for this architecture are the time-window size ($w$), the ovelap ratio, the size and number of filters in each of the convolutional layers , and the size of fully connected layer before the logit layer. The number of layers depends on the size of the input window.”
- what's the size of each CNN layer, what's the input dimension of images that are fed, etc. Is N=24 in their figure, etc.?
Changed the TCNN architecture description: “The number of convolutional layers for the Fantasia and MIT-BIH databases is 6 of sizes 256, 128, 64, 16, 8, and 4 for the non-fiducial network and 4 of 64, 16, 8, and 4 for the fiducial network. The input window was of 256 samples for these databases. For the CYBHi database, due to the increase of the sampling frequency, each network had an increase of two initial layers, 1024 and 512 added to the previous layers of the non-fiducial network, and 256 and 128 added to the previous layers of the fiducial network. The input window of this dataset had 1028 samples.”
Overall, I think there is significant scope of improvement in paper as suggested above before it can be published.
Minor corrections (for example):
- "Both outputs were 7 to the same classifier, the Relative Score Threshold Classifier (RSTC) which exploits the error of 8 prediction of the former and the logits given by the last. "
This sentence is not very clear. Can the authors rephrase?
- "Both scores are submitted the Relative Score Threshold Classifier (RSTC)"
Missing preposition "to", also perhaps wrong usage of word "submitted", maybe fed into is more appropriate?
- both methods will be tested "in" three?
should it be with?
- "Recently, research in ECG biometry has been using DNN"
- "The main advantages of the RNN architecture are the is that all the models a"
and many more ...
Thank you so much for pointing out these improvement points as they made this article much better if any are added please inform us so we can make the required changes. All the text was reviewed and re-written and the informed minor corrections were made.
Reviewer 2 Report
Pros:
- The authors present a new methodology that incorporates DNNs for extracting features for biometric identification. This seems to be an innovative approach and has some potential to lead to good results as demonstrated in the results section.
Cons:
- The introduction does not seem to have enough citations. Incorporate more relevant background describing applications of DNNs to ECG processing would be helfpul. Why not use/modify one of the existing architectures for this problem?
- The methodology (starting with the problem description) is not well described. It is not clear how they defined the problem from the beginning. There is some hints on the results section but it would be helpful to have a clear problem statement to begin with.
- Some variables such as s_n are never described, and some of the notation is not appropriate. For example, equation 4 seems to define a variable using the exact same notation as another variable.
- Details on how the networks were trained are lacking. At some point, the authors just mention a prediction error, but it is not clear what they are referring to.
- There is no justification why the specified network structures were selected. Was this the result of some hyper-parameter and structure optimization / search? If so, how was this done? Why not use other architectures inspired by the literature of DNNs applied to ECG?
- The result section seem clear but it was not hard to really interpret some of the results since the methodology was not clearly stated.
- Table 1 shows that the approaches did not do as well as reference [3] when evaluating using the "C M1-M2" dataset. The authors should expand on this. Does this show that their method does not generalize well in this case? Would this be solved if they had more data?
Author Response
Thank you for reviewing this article, the authors analyzed all comments and incorporated them to improve the quality of the paper significantly. We ask for forgiveness in the English as we are not native speakers but we put an effort to enhance the English quality while re-writing the article. Each bullet point was addressed as follows:
Pros:
- The authors present a new methodology that incorporates DNNs for extracting features for biometric identification. This seems to be an innovative approach and has some potential to lead to good results as demonstrated in the results section.
Cons:
- The introduction does not seem to have enough citations. Incorporate more relevant background describing applications of DNNs to ECG processing would be helfpul. Why not use/modify one of the existing architectures for this problem?
Even though there are studies in the ECG biometry which use DNN architectures, there aren’t much that we are aware of. Even though we added several entries of these systems and improved the “Related Work” first paragraphs, reassuring the need to improve these methods.
“Recently, ECG biometry researchers have been using DNN, in which, most architectures comprise Convolutional Neural Networks (CNN) modules that both process and classify. The inputs to these networks may include the human-extracted features such as the temporal frequency spectrograms obtained by discrete wavelet transform in [20] or [21] which incorporates time-based features and uses a feed-forward network for classification. In [22] a feed-forward network is used for feature extraction and classification. The authors in [23] and [24] explores the use of CNN for both authentication and authentication of datasets for feature extraction and classification, but before feeding the signal to the network an average wave is calculated, surrounding the R peak of the ECG signal. The systems developed by [25] and [26] also used DNN but they will be discussed later as they use the CYBHi and MIT-BIH datasets, respectively.
This work proposes the improvement of these methods with the generalization of the abstract notion of an individualized ECG by using a non-fiducial and semi-fiducial systems and using time-series, instead of the feature extraction methods or resorting to the transformation of the signal to an image. It will also be tested in an off-person database which as high intra-variability between subjects. For benchmarking purposes, the rest of this section will address related work using the referenced databases: Fantasia, MIT-BIH, and CYBHi database.
Changes in the “MIT-BIH” subsection of the “Related Work” section:
“The work developed by [26] claim to achieve a 98.55 % of accuracy in identification paradigm using a bi-directional Gated Recurrent Units (GRU) network model incorporating for 47 records of the MIT-BIH Arrhythmia database. In this article, they compare several architectures, including CNN and Long Short-Term Memory (LSTM) models with a fiducial approach.”
- The methodology (starting with the problem description) is not well described. It is not clear how they defined the problem from the beginning. There is some hints on the results section but it would be helpful to have a clear problem statement to begin with.
The authors agree strongly with this statement and therefore added information on the description of the problem and the addressing of those issues regarding the noise and intra-variability. To address the pointed issue changes to the introduction were made mentioning the types of noise, the difficulties that the biometrics challenge, and the reasoning behind the use of DNN.
Changes in the “ECG biometrics” subsection of the “Introduction:
“Given its individual morphological signature and advances in sensoring devices, the ECG emerged in the last decade as a biometric modality with the promise of robustness against circumvention attacks, and ability to continuous pervasive acquisition scenarios. The main difficulties that involve the non-invasive ECG measurements are the intra-subject variability, artifacts, and noise [10,11] making the computation of the features arduous, particularly when the ECG signal is contaminated in the characteristic points of the waveform [12]. Merone et al. (2017)[13] compiled several sources of artifacts and noise, such as, electrode material, sensor locations, power-line interference, movement artifacts and instrumentation of the devices. As for the intra-subject variability, it may reside in the health status of the patient, the heart-rate variability, physical exercise, affective status, and drugs, but Wübbeler et al. (2007)[14] states that the ECG remains stable over the years, the sufficient to allow recognition with an error rate of 3% [10,11,14].“
Changes in the “Objectives” subsection of the “Introduction:
“The issue of the intra-variability in each subject which exists in real-life scenarios, ECG biometric systems struggle to find their way outside controlled settings. This work has the objective to provide one step further the abstraction of the individualized ECG signal to be used in real-life scenarios using state-of-the-art technologies such as DNN.
Two architectures are proposed as an an effort to improve performance in both identification and authentication without extracting human-crafted features (Fig. \ref{fig:biometry_architectures}): (1) a non-fiducial system that uses a \gls{RNN} capable synthesizing ECG signals, outputting a score based on the error of prediction; (2) a partial-fiducial using a TCNN, giving a score based on the output of the last layer. Both scores are fed to the RSTC which classifies the given sample or a batch of samples.”
- Some variables such as s_n are never described, and some of the notation is not appropriate. For example, equation 4 seems to define a variable using the exact same notation as another variable.
The authors thank the reviewer for this comment, the s_n as been described. The equation 4 the symbols are just as intended, but the brief description was made to make it clear: “where the notation B.b represents the multiplication of the batch number by the batch size.” If b was not the concern variable, please be so kind to state which one so we can correct it.
- Details on how the networks were trained are lacking. At some point, the authors just mention a prediction error, but it is not clear what they are referring to.
The following information was added to the “Methods and Materials” section:
The RNN subsection:
“These models were trained using mean-squared error using the RMSProp [35] optimization algorithm. The prediction error, i.e. loss function, is given by the following equation:
ep=(1/W) n=0W-1(yn−y)2 (3)
where the W is the window size, ^yn is the predicted n-th sample and yn the real sample value.”
The TCNN subsection:
“Each CNN are trained using the Adam optimizer [37] and the loss function (L) is given by the cross-entropy error:
L=−(1/I) nI [i log(ˆi)+(1−i)log(i−î)] (4)
where the i is the number of subjects window size,î is the predicted subject number, while i is the real subject number.
- There is no justification why the specified network structures were selected. Was this the result of some hyper-parameter and structure optimization / search? If so, how was this done? Why not use other architectures inspired by the literature of DNNs applied to ECG?
The following information was added to the “Methods and Materials” section:
The RNN subsection:
“The synthesizing of ECG signals in the previous work [34] provided a hypothesis that if the model that is trained with signal A and fed with signal B from another individual, the prediction error will be higher than if fed with a signal with the original individual A. After some testing and observation of the results, it was confirmed that such premise could be explored. The hyper-parameter search was made through careful observation and continuous try-and-error procedures in search for better results using a subset of the Fantasia dataset.”
The TCNN subsection:
“Earlier configurations of this CNN were made and tested including the input of spectrograms [37] and 1-CNN without the use of dilated convolutions. After testing using Fantasia and CYBHi datasets with try-and-error parameter hyper-tuning, the resultant architecture provided the best results of the previous CNN versions.”
- The result section seem clear but it was not hard to really interpret some of the results since the methodology was not clearly stated.
Thank you so much for this comment. Since the methodology section was modified, regarding these comments and the reviews from other reviewers, we hope that this concern was cleared. If not, please help us by stating which sections of the methodology are not clear so we can explain them better.
- Table 1 shows that the approaches did not do as well as reference [3] when evaluating using the "C M1-M2" dataset. The authors should expand on this. Does this show that their method does not generalize well in this case? Would this be solved if they had more data?
In truth, the case of “C M1-M2” dataset for [3], is that they only use a subset of the subjects, while we used all records. Therefore we changed the first paragraphs of the “Discussion” section:
“Table 1 summarizes the results per database for the consulted bibliography. This table shows that he TCNN approach outperforms most of the other studies in both identification and authentication paradigms. The robustness of this architecture is mainly due to the combination of bringing together both fiducial and non-fiducial machine-learned characteristics of the signal through the fusion layer. The only exception was for the CYBHi M1 vs M2 results for the identification modality, as Lourenço et al. (2012) [5] has higher results, but it can be explained by two reasons: (1) only 32, versus the 68, people were used; (2) the procedure was completely fiducial-based.
It is possible to observe that the results between moments in the CYBHi dataset are lower due, not only to the increase of the noise and artifacts displayed in the signal but also to the changes in the mental and emotional states of each user. This statement enforces the challenge that the biometrics systems face due to the intra-variability of the subjects. Therefore more acquisition setups should be made, such as increasing the number of people, increasing the acquisition time, and increasing the number of moments of acquisition. The amount of cleaner data would impact significantly the training of the models. Consequently, these measures would not only provide higher accuracy and EER but also more confidence in the shown results to claim that the systems were capable of a successful generalization.”
Round 2
Reviewer 1 Report
I'm mostly satisfied with the changes made by the authors except for one particular point with regards to the model accuracy/error.
Authors do mention that the computational power was far too low for changes in the variation of the cross-validation and boot-strapping. Also, they mention "Each measurement usually takes weeks to make (due to training), and some variations could take months to print the results."
For scalability of any system at industrial scale, both training (particularly retraining) time/inference play a significant role. At the same time, I do agree that not all work can be done as part of a single scientific research, and some work ought to be left for future work.
I would suggest the authors to comment on training time/inference time of their method with other methods out there. A rigorous analysis may not be necessary but those reading the paper should be aware of what the "extra" accuracy/error is coming at cost of.
Finally, quoting "100%" accuracy without any confidence intervals on the output looks a bit misleading. I would suggest the authors to add a disclaimer with regards to accuracy/error particularly for Fantasia and/or other datasets. Perhaps a better way to comment on their improvement is to mention that improvement is say x% improvement with current state of art methods or to say close to 100% (instead of saying 100%) since authors really don't have a confidence interval or a bound to it.
For conclusion, I would suggest authors to add one more point that is worth mentioning: to improve the "training/retraining/inference" time. Most current industrial biometric systems have close to real time performance, and are easily extendable when more subjects are added to the system.
Author Response
I'm mostly satisfied with the changes made by the authors except for one particular point with regards to the model accuracy/error.
Thank you so much for your contribution, it was paramount for this outcome.
Authors do mention that the computational power was far too low for changes in the variation of the cross-validation and boot-strapping. Also, they mention "Each measurement usually takes weeks to make (due to training), and some variations could take months to print the results."
For scalability of any system at industrial scale, both training (particularly retraining) time/inference play a significant role. At the same time, I do agree that not all work can be done as part of a single scientific research, and some work ought to be left for future work.
I would suggest the authors to comment on training time/inference time of their method with other methods out there. A rigorous analysis may not be necessary but those reading the paper should be aware of what the "extra" accuracy/error is coming at cost of.
The authors totally agree with this statement. Therefore several approaches were made. Changes made in the Discussion:
- “The high accuracy that both approaches obtained comes with a cost on the training time. For example, the RNN training time, using the CYBHi database and a computer with a nVidia GTX 1080Ti GPU, take an average of 36 hours per subject, totaling approximately 2300 hours for this database. The implementation of this system at an industrial level this limitation should be mitigated as retraining should be faster, especially when the number of individuals increases.“ was added.
- we also take this opportunity to mention that later we talk about this limitation and give one possible solution: “The implementation of a biometric system including the \gls{TCNN} approach in a real-life scenario recognizes several challenges: (1) the high computational time, which increases significantly with the number of samples per window and the number of individuals (...)
For tackling these concerns, the use of transfer learning could prove useful. If the training of a network introduced a first learning stage with a different database for recognition of the basic structures and inner mechanisms of the ECG morphology, the second stage of learning, it would fine-tune the appropriate filters for source recognition and optimize faster.”
Finally, quoting "100%" accuracy without any confidence intervals on the output looks a bit misleading. I would suggest the authors to add a disclaimer with regards to accuracy/error particularly for Fantasia and/or other datasets. Perhaps a better way to comment on their improvement is to mention that improvement is say x% improvement with current state of art methods or to say close to 100% (instead of saying 100%) since authors really don't have a confidence interval or a bound to it.
We aim to be the most transparent possible, therefore every time we mention “100%” we say “close to” or “almost”.
For conclusion, I would suggest authors to add one more point that is worth mentioning: to improve the "training/retraining/inference" time. Most current industrial biometric systems have close to real time performance, and are easily extendable when more subjects are added to the system.
We understand that this may become a major issue to apply these solutions to the real world. Consequently, we added this comment to the conclusions to reaffirm the importance of this fact: “Finally, the training/retraining/inference time of these systems should be analyzed and enhanced for real-life applications for delivering close to real-time solutions.”
Reviewer 2 Report
The authors have improved the article and addressed several of the issues originally pointed out. However, there are still some issues that need to be resolved.
- Please have someone take a look at the entire document for grammar and formatting issues.
- Paragraph starting at line 150 talks about "moment" but I believe the authors mean "session"
- The sentence starting in line 153 needs to be capitalized
- Lines after equation should not be indented and sometimes they start with a comma when the comma should be in the same line as the equation
- The hat on top of y in equation (3) should be only applied to the y (i.e., as in \hat{y} instead of \hat{y_n})
- s_n in equation (1) has not been defined
- The use of e_p in equation (2) is not clear. I believe e(p,i,w) should be used instead. Similar comment for the variable o_p in equation (5)
- Line 197 introduces W when it may just be w
- In section 2.3, it is not clear what response is used for training of the network. Is it y_n = s_n?
- Line 233, what is M?
- For equation (6), it would be good to use \tilde{S}_{p,i,b} instead of S_{p,i,b} to avoid reusing the same variable. This change will require some updates for equation (7)
- Line 248, instead of phi, I believe the authors meant to use delta
- Equation (9) is not clear
- Line 263 refers to the subplot as been to the "left" and "right" when they are on top and bottom in Figure 4.
- In line 295, it is said that the RNN does not go above 80% but Fig. 7(a) seems to show that they get to be above 90%.
Author Response
The authors have improved the article and addressed several of the issues originally pointed out. However, there are still some issues that need to be resolved.
Thank you so much for your contribution, it was paramount for this outcome. All the pointed issues were dealt with, and while we made the text revision (after a dozen cycles) we decided to check the text paragraph by paragraph, and more typos and “copy-paste” mistakes were corrected. We also made the text more clear and readable for the audience. We truly do hope we did a good job.
Please have someone take a look at the entire document for grammar and formatting issues.
Paragraph starting at line 150 talks about "moment" but I believe the authors mean "session"
Changed all “moments” to “sessions”. It was a direct translation from Portuguese, we do realize “session” is a better word to describe it.
The sentence starting in line 153 needs to be capitalized
Lines after equation should not be indented and sometimes they start with a comma when the comma should be in the same line as the equation
The hat on top of y in equation (3) should be only applied to the y (i.e., as in \hat{y} instead of \hat{y_n})
All the above changes were made.
s_n in equation (1) has not been defined
It is now clear. We changed the text after equation (1) “where $x_n$ is the $n$-th sample of the input vector and $s_n$ is the raw $n$-th sample of the raw signal ($s$).” and repeat after equation (3): “Note that the $s_n$ is the raw signal (...).
The use of e_p in equation (2) is not clear. I believe e(p,i,w) should be used instead. Similar comment for the variable o_p in equation (5)
We agree with this comment and changed accordingly.
Line 197 introduces W when it may just be w
Actually “W” is the window size and “w” is the index of the window. We realized that they were switched in some places of the text and equations and, therefore, it was corrected in all references of “W” and “w”.
In section 2.3, it is not clear what response is used for training of the network. Is it y_n = s_n?
Thank you for pointing out this issue. We added after that equation: “where the $W$ is the window size (number of samples in the time-window), $\hat{y}_n$ the predicted $n$-th sample and $y_n$ the real sample value. Note that the $s_n$ is the raw signal, $x_n$ is the quantized signal, while $y_n$ is equal to $x_n$ dephased by one sample since the \gls{RNN} architecture predicts the next sample \cite{Belo2017}.”
We would like to mention that we also added a better explanation of the training process of this network: “Each model is trained for each subject while predicting the next sample amplitude.”
Line 233, what is M?
Removed, we realized it was not needed, as M (number of predictors) was equal to I (number of individuals) so we changed the matrix dimensions from “(M x I x B)” to “(I x I x B)“
For equation (6), it would be good to use \tilde{S}_{p,i,b} instead of S_{p,i,b} to avoid reusing the same variable. This change will require some updates for equation (7)
Line 248, instead of phi, I believe the authors meant to use delta
All the above changes were made.
Equation (9) is not clear
We transformed the equation to a much clearer system.
C = 1, if Ì„S(p,i,b)≤δ
C = 0, if Ì„S(p,i,b)>δ
"when C has the value 1 the classification for class A is positive, while 0 when negative." (please check the article directly as we cannot make equations in this form)
Line 263 refers to the subplot as been to the "left" and "right" when they are on top and bottom in Figure 4.
Changed to 4a and 4b, instead of left and right.
In line 295, it is said that the RNN does not go above 80% but Fig. 7(a) seems to show that they get to be above 90%.
We changed all the paragraph mentioning this.